# Consistency of weighted majority votes

**Daniel Berend Computer Science Department and Mathematics Department**
Ben Gurion University
Beer Sheva, Israel `berend@cs.bgu.ac.il`

**Aryeh Kontorovich**
Computer Science Department Ben Gurion University
Beer Sheva, Israel `karyeh@cs.bgu.ac.il`

## Abstract

We revisit from a statistical learning perspective the classical decision-theoretic problem of weighted expert voting. In particular, we examine the consistency (both asymptotic and finitary) of the optimal Nitzan-Paroush weighted majority and related rules. In the case of known expert competence levels, we give sharp error estimates for the optimal rule. When the competence levels are unknown, they must be empirically estimated. We provide frequentist and Bayesian analyses for this situation. Some of our proof techniques are non-standard and may be of independent interest. The bounds we derive are nearly optimal, and several challenging open problems are posed.

## 1 Introduction

Imagine independently consulting a small set of medical experts for the purpose of reaching a binary decision (e.g., whether to perform some operation). Each doctor has some "reputation", which can be modeled as his probability of giving the right advice. The problem of weighting the input of several experts arises in many situations and is of considerable theoretical and practical importance. The rigorous study of majority vote has its roots in the work of Condorcet [1]. By the 70s, the field of decision theory was actively exploring various voting rules (see [2] and the references therein). A typical setting is as follows. An agent is tasked with predicting some random variable $Y \in \{\pm 1\}$ based on input $X_i \in \{\pm 1\}$ from each of $n$ experts. Each expert $X_i$ has a *competence* level $p_i \in (0,1)$, which is the probability of making a correct prediction: $\mathbb{P}(X_i = Y) = p_i$. Two simplifying assumptions are commonly made:

(i) *Independence*: The random variables $\{X_i : i \in [n]\}$ are mutually independent conditioned on the truth $Y$.

(ii) *Unbiased truth*: $\mathbb{P}(Y = +1) = \mathbb{P}(Y = -1) = 1/2$.

We will discuss these assumptions below in greater detail; for now, let us just take them as given. (Since the bias of $Y$ can be easily estimated from data, only the independence assumption is truly restrictive.) A *decision rule* is a mapping $f : \{\pm 1\}^n \to \{\pm 1\}$ from the $n$ expert inputs to the agent's final decision. Our quantity of interest throughout the paper will be the agent's probability of error,

$$\mathbb{P}(f(\mathbf{X}) \neq Y). \tag{1}$$

A decision rule $f$ is *optimal* if it minimizes the quantity in (1) over all possible decision rules. It was shown in [2] that, when Assumptions (i)–(ii) hold and the true competences $p_i$ are known, the optimal decision rule is obtained by an appropriately weighted majority vote:

$$f^{\mathrm{OPT}}(\mathbf{x}) = \mathrm{sign}\left(\sum_{i=1}^{n} w_i x_i\right), \tag{2}$$

where the weights $w_i$ are given by

$$w_i = \log \frac{p_i}{1 - p_i}, \qquad i \in [n]. \tag{3}$$

Thus, $w_i$ is the log-odds of expert $i$ being correct — and the voting rule in (2), also known as *naive Bayes* [3], may be seen as a simple consequence of the Neyman-Pearson lemma [4].

**Main results.** The formula in (2) raises immediate questions, which apparently have not previously been addressed. The first one has to do with the *consistency* of the Nitzan-Paroush optimal rule: under what conditions does the probability of error decay to zero and at what rate? In Section 3, we show that the probability of error is controlled by the *committee potential* $\Phi$, defined by

$$\Phi = \sum_{i=1}^{n} (p_i - \tfrac{1}{2}) w_i = \sum_{i=1}^{n} (p_i - \tfrac{1}{2}) \log \frac{p_i}{1 - p_i}. \tag{4}$$

More precisely, we prove in Theorem 1 that $\log \mathbb{P}(f^{\mathrm{OPT}}(\mathbf{X}) \neq Y) \asymp -\Phi$, where $\asymp$ denotes equivalence up to universal multiplicative constants.

Another issue not addressed by the Nitzan-Paroush result is how to handle the case where the competences $p_i$ are not known exactly but rather estimated empirically by $\hat{p}_i$. We present two solutions to this problem: a frequentist and a Bayesian one. As we show in Section 4, the frequentist approach does not admit an optimal empirical decision rule. Instead, we analyze empirical decision rules in various settings: high-confidence (i.e., $|\hat{p}_i - p_i| \ll 1$) vs. low-confidence, adaptive vs. nonadaptive. The low-confidence regime requires no additional assumptions, but gives weaker guarantees (Theorem 5). In the high-confidence regime, the adaptive approach produces error estimates in terms of the empirical $\hat{p}_i$s (Theorem 7), while the nonadaptive approach yields a bound in terms of the unknown $p_i$s, which still leads to useful asymptotics (Theorem 6). The Bayesian solution sidesteps the various cases above, as it admits a simple, provably optimal empirical decision rule (Section 5). Unfortunately, we are unable to compute (or even nontrivially estimate) the probability of error induced by this rule; this is posed as a challenging open problem.

## 2 Related work

Machine learning theory typically clusters *weighted majority* [5, 6] within the framework of online algorithms; see [7] for a modern treatment. Since the online setting is considerably more adversarial than ours, we obtain very different weighted majority rules and consistency guarantees. The weights $w_i$ in (2) bear a striking similarity to the Adaboost update rule [8, 9]. However, the latter assumes weak learners with access to labeled examples, while in our setting the experts are "static". Still, we do not rule out a possible deeper connection between the Nitzan-Paroush decision rule and boosting.

In what began as the influential Dawid-Skene model [10] and is now known as *crowdsourcing*, one attempts to extract accurate predictions by pooling a large number of experts, typically without the benefit of being able to test any given expert's competence level. Still, under mild assumptions it is possible to efficiently recover the expert competences to a high accuracy and to aggregate them effectively [11]. Error bounds for the oracle MAP rule were obtained in this model by [12] and minimax rates were given in [13].

In a recent line of work [14, 15, 16] have developed a PAC-Bayesian theory for the majority vote of simple classifiers. This approach facilitates data-dependent bounds and is even flexible enough to capture some simple dependencies among the classifiers — though, again, the latter are *learners* as opposed to our *experts*. Even more recently, experts with adversarial noise have been considered [17], and efficient algorithms for computing optimal expert weights (without error analysis) were given [18]. More directly related to the present work are the papers of [19], which characterizes the consistency of the simple majority rule, and [20, 21, 22] which analyze various models of dependence among the experts.

## 3 Known competences

In this section we assume that the expert competences $p_i$ are known and analyze the consistency of the Nitzan-Paroush optimal decision rule (2). Our main result here is that the probability of error $\mathbb{P}(f^{\text{OPT}}(\mathbf{X}) \neq Y)$ is small if and only if the committee potential $\Phi$ is large.

**Theorem 1.** *Suppose that the experts* $\mathbf{X} = (X_1, \ldots, X_n)$ *satisfy Assumptions (i)-(ii) and* $f^{\text{OPT}} : \{\pm 1\}^n \to \{\pm 1\}$ *is the Nitzan-Paroush optimal decision rule. Then*

*(i)* $\mathbb{P}(f^{\text{OPT}}(\mathbf{X}) \neq Y) \leq \exp\left(-\frac{1}{2}\Phi\right)$.

*(ii)* $\mathbb{P}(f^{\text{OPT}}(\mathbf{X}) \neq Y) \geq \dfrac{3}{8[1 + \exp(2\Phi + 4\sqrt{\Phi})]}$.

As we show in the full paper [27], the upper and lower bounds are both asymptotically tight. The remainder of this section is devoted to proving Theorem 1.

### 3.1 Proof of Theorem 1(i)

Define the $\{0, 1\}$-indicator variables

$$\xi_i = \mathbb{1}_{\{X_i = Y\}}, \tag{5}$$

corresponding to the event that the $i^{\text{th}}$ expert is correct. A mistake $f^{\text{OPT}}(\mathbf{X}) \neq Y$ occurs precisely when[1] the sum of the correct experts' weights fails to exceed half the total mass:

$$\mathbb{P}(f^{\text{OPT}}(\mathbf{X}) \neq Y) = \mathbb{P}\left(\sum_{i=1}^{n} w_i \xi_i \leq \frac{1}{2}\sum_{i=1}^{n} w_i\right). \tag{6}$$

Since $\mathbb{E}\xi_i = p_i$, we may rewrite the probability in (6) as

$$\mathbb{P}\left(\sum_i w_i \xi_i \leq \mathbb{E}\left[\sum_i w_i \xi_i\right] - \sum_i (p_i - \tfrac{1}{2})w_i\right). \tag{7}$$

A standard tool for estimating such sum deviation probabilities is Hoeffding's inequality. Applied to (7), it yields the bound

$$\mathbb{P}(f^{\text{OPT}}(\mathbf{X}) \neq Y) \leq \exp\left(-\frac{2\left[\sum_i (p_i - \tfrac{1}{2})w_i\right]^2}{\sum_i w_i^2}\right), \tag{8}$$

which is far too crude for our purposes. Indeed, consider a finite committee of highly competent experts with $p_i$'s arbitrarily close to 1 and $X_1$ the most competent of all. Raising $X_1$'s competence sufficiently far above his peers will cause both the numerator and the denominator in the exponent to be dominated by $w_1^2$, making the right-hand-side of (8) bounded away from zero. The inability of Hoeffding's inequality to guarantee consistency even in such a felicitous setting is an instance of its generally poor applicability to highly heterogeneous sums, a phenomenon explored in some depth in [23]. Bernstein's and Bennett's inequalities suffer from a similar weakness (see ibid.). Fortunately, an inequality of Kearns and Saul [24] is sufficiently sharp to yield the desired estimate: For all $p \in [0, 1]$ and all $t \in \mathbb{R}$,

$$(1-p)e^{-tp} + pe^{t(1-p)} \leq \exp\left(\frac{1-2p}{4\log((1-p)/p)}t^2\right). \tag{9}$$

**Remark.** *The Kearns-Saul inequality (9) may be seen as a distribution-dependent refinement of Hoeffding's (which bounds the left-hand-side of (9) by* $e^{t^2/8}$*), and is not nearly as straightforward to prove. An elementary rigorous proof is given in [25]. Following up, [26] gave a "soft" proof based on transportation and information-theoretic techniques.*

Put $\theta_i = \xi_i - p_i$, substitute into (6), and apply Markov's inequality:

$$\mathbb{P}(f^{\text{OPT}}(\mathbf{X}) \neq Y) = \mathbb{P}\left(-\sum_i w_i\theta_i \geq \Phi\right) \leq e^{-t\Phi}\mathbb{E}\exp\left(-t\sum_i w_i\theta_i\right). \tag{10}$$

Now

$$\begin{aligned}
\mathbb{E}e^{-tw_i\theta_i} &= p_i e^{-(1-p_i)w_i t} + (1-p_i)e^{p_i w_i t} \\
&\leq \exp\left(\frac{-1+2p_i}{4\log(p_i/(1-p_i))}w_i^2 t^2\right) = \exp\left[\tfrac{1}{2}(p_i - \tfrac{1}{2})w_i t^2\right],
\end{aligned} \tag{11}$$

where the inequality follows from (9). By independence,

$$\mathbb{E}\exp\left(-t\sum_i w_i\theta_i\right) = \prod_i \mathbb{E}e^{-tw_i\theta_i} \leq \exp\left(\tfrac{1}{2}\sum_i (p_i - \tfrac{1}{2})w_i t^2\right) = \exp\left(\tfrac{1}{2}\Phi t^2\right)$$

and hence $\mathbb{P}(f^{\text{OPT}}(\mathbf{X}) \neq Y) \leq \exp\left(\tfrac{1}{2}\Phi t^2 - \Phi t\right)$. Choosing $t = 1$ yields the bound in Theorem 1(i).

### 3.2 Proof of Theorem 1(ii)

Define the $\{\pm 1\}$-indicator variables

$$\eta_i = 2 \cdot \mathbb{1}_{\{X_i = Y\}} - 1, \tag{12}$$

corresponding to the event that the $i^{\text{th}}$ expert is correct and put $q_i = 1 - p_i$. The shorthand $\mathbf{w} \cdot \boldsymbol{\eta} = \sum_{i=1}^n w_i\eta_i$ will be convenient. We will need some simple lemmata, whose proofs are deferred to the journal version [27].

**Lemma 2.**

$$\mathbb{P}(f^{\text{OPT}}(\mathbf{X}) = Y) = \tfrac{1}{2}\sum_{\boldsymbol{\eta}\in\{\pm 1\}^n} \max\{P(\boldsymbol{\eta}), P(-\boldsymbol{\eta})\}$$

*and*

$$\mathbb{P}(f^{\text{OPT}}(\mathbf{X}) \neq Y) = \tfrac{1}{2}\sum_{\boldsymbol{\eta}\in\{\pm 1\}^n} \min\{P(\boldsymbol{\eta}), P(-\boldsymbol{\eta})\},$$

*where* $P(\boldsymbol{\eta}) = \prod_{i:\eta_i=1} p_i \prod_{i:\eta_i=-1} q_i$.

**Lemma 3.** *Suppose that* $\mathbf{s}, \mathbf{s}' \in (0,\infty)^m$ *satisfy* $\sum_{i=1}^m (s_i + s_i') \geq a$ *and* $R^{-1} \leq s_i/s_i' \leq R$, $i \in [m]$, *for some* $R < \infty$. *Then* $\sum_{i=1}^m \min\{s_i, s_i'\} \geq a/(1+R)$.

**Lemma 4.** *Define the function* $F : (0,1) \to \mathbb{R}$ *by*

$$F(x) = \frac{x(1-x)\log(x/(1-x))}{2x - 1}.$$

*Then* $\sup_{0<x<1} F(x) = \tfrac{1}{2}$.

Continuing with the main proof, observe that

$$\mathbb{E}[\mathbf{w} \cdot \boldsymbol{\eta}] = \sum_{i=1}^n (p_i - q_i)w_i = 2\Phi \tag{13}$$

and $\text{Var}[\mathbf{w} \cdot \boldsymbol{\eta}] = 4\sum_{i=1}^n p_i q_i w_i^2$. By Lemma 4, $p_i q_i w_i^2 \leq \tfrac{1}{2}(p_i - q_i)w_i$, and hence

$$\text{Var}[\mathbf{w} \cdot \boldsymbol{\eta}] \leq 4\Phi. \tag{14}$$

Define the segment $I \subset \mathbb{R}$ by

$$I = \left[2\Phi - 4\sqrt{\Phi}, 2\Phi + 4\sqrt{\Phi}\right]. \tag{15}$$

Chebyshev's inequality together with (13) and (14) implies that

$$\mathbb{P}(\mathbf{w} \cdot \boldsymbol{\eta} \in I) \geq \frac{3}{4}. \tag{16}$$

Consider an atom $\boldsymbol{\eta} \in \{\pm 1\}^n$ for which $\mathbf{w} \cdot \boldsymbol{\eta} \in I$. The proof of Lemma 2 shows that

$$\frac{P(\boldsymbol{\eta})}{P(-\boldsymbol{\eta})} = \exp\left(\mathbf{w} \cdot \boldsymbol{\eta}\right) \le \exp(2\Phi + 4\sqrt{\Phi}), \tag{17}$$

where the inequality follows from (15). Lemma 2 further implies that

$$\mathbb{P}(f^{\mathrm{OPT}}(\mathbf{X}) \ne Y) \ge \tfrac{1}{2} \sum_{\boldsymbol{\eta} \in \{\pm 1\}^n : \mathbf{w} \cdot \boldsymbol{\eta} \in I} \min\left\{P(\boldsymbol{\eta}), P(-\boldsymbol{\eta})\right\} \ge \frac{3/4}{1 + \exp(2\Phi + 4\sqrt{\Phi})},$$

where the second inequality follows from Lemma 3, (16) and (17). This completes the proof.

# 4  Unknown competences: frequentist

Our goal in this section is to obtain, insofar as possible, analogues of Theorem 1 for unknown expert competences. When the $p_i$s are unknown, they must be estimated empirically before any useful weighted majority vote can be applied. There are various ways to model partial knowledge of expert competences [28, 29]. Perhaps the simplest scenario for estimating the $p_i$s is to assume that the $i^{\mathrm{th}}$ expert has been queried independently $m_i$ times, out of which he gave the correct prediction $k_i$ times. Taking the $\{m_i\}$ to be fixed, define the *committee profile* by $\mathbf{k} = (k_1, \ldots, k_n)$; this is the aggregate of the agent's empirical knowledge of the experts' performance. An *empirical decision rule* $\hat{f} : (\mathbf{x}, \mathbf{k}) \mapsto \{\pm 1\}$ makes a final decision based on the expert inputs $\mathbf{x}$ together with the committee profile. Analogously to (1), the probability of a mistake is

$$\mathbb{P}(\hat{f}(\mathbf{X}, \mathbf{K}) \ne Y). \tag{18}$$

Note that now the committee profile is an additional source of randomness. Here we run into our first difficulty: unlike the probability in (1), which is minimized by the Nitzan-Paroush rule, the agent cannot formulate an optimal decision rule $\hat{f}$ in advance without knowing the $p_i$s. This is because no decision rule is optimal uniformly over the range of possible $p_i$s. Our approach will be to consider weighted majority decision rules of the form

$$\hat{f}(\mathbf{x}, \mathbf{k}) = \mathrm{sign}\left(\sum_{i=1}^{n} \hat{w}(k_i) x_i\right) \tag{19}$$

and to analyze their consistency properties under two different regimes: low-confidence and high-confidence. These refer to the confidence intervals of the frequentist estimate of $p_i$, given by

$$\hat{p}_i = \frac{k_i}{m_i}. \tag{20}$$

## 4.1  Low-confidence regime

In the low-confidence regime, the sample sizes $m_i$ may be as small as 1, and we define[2]

$$\hat{w}(k_i) = \hat{w}_i^{\mathrm{LC}} := \hat{p}_i - \tfrac{1}{2}, \qquad i \in [n], \tag{21}$$

which induces the empirical decision rule $\hat{f}^{\mathrm{LC}}$. It remains to analyze $\hat{f}^{\mathrm{LC}}$'s probability of error. Recall the definition of $\xi_i$ from (5) and observe that

$$\mathbb{E}\left[\hat{w}_i^{\mathrm{LC}} \xi_i\right] = \mathbb{E}[(\hat{p}_i - \tfrac{1}{2})\xi_i] = (p_i - \tfrac{1}{2})p_i, \tag{22}$$

since $\hat{p}_i$ and $\xi_i$ are independent. As in (6), the probability of error (18) is

$$\mathbb{P}\left(\sum_{i=1}^{n} \hat{w}_i^{\mathrm{LC}} \xi_i \le \frac{1}{2} \sum_{i=1}^{n} \hat{w}_i^{\mathrm{LC}}\right) = \mathbb{P}\left(\sum_{i=1}^{n} Z_i \le 0\right), \tag{23}$$

where $Z_i = \hat{w}_i^{\text{LC}}(\xi_i - \frac{1}{2})$. Now the $\{Z_i\}$ are independent random variables, $\mathbb{E}Z_i = (p_i - \frac{1}{2})^2$ (by (22)), and each $Z_i$ takes values in an interval of length $\frac{1}{2}$. Hence, the standard Hoeffding bound applies:

$$\mathbb{P}(\hat{f}^{\text{LC}}(\mathbf{X}, \mathbf{K}) \neq Y) \leq \exp\left[-\frac{8}{n}\left(\sum_{i=1}^{n}(p_i - \tfrac{1}{2})^2\right)^2\right]. \tag{24}$$

We summarize these calculations in

**Theorem 5.** *A sufficient condition for* $\mathbb{P}(\hat{f}^{\text{LC}}(\mathbf{X}, \mathbf{K}) \neq Y) \to 0$ *is* $\frac{1}{\sqrt{n}}\sum_{i=1}^{n}(p_i - \frac{1}{2})^2 \to \infty$.

Several remarks are in order. First, notice that the error bound in (24) is stated in terms of the unknown $\{p_i\}$, providing the agent with large-committee asymptotics but giving no finitary information; this limitation is inherent in the low-confidence regime. Secondly, the condition in Theorem 5 is considerably more restrictive than the consistency condition $\Phi \to \infty$ implicit in Theorem 1. Indeed, the empirical decision rule $\hat{f}^{\text{LC}}$ is incapable of exploiting a single highly competent expert in the way that $f^{\text{OPT}}$ from (2) does. Our analysis could be sharpened somewhat for moderate sample sizes $\{m_i\}$ by using Bernstein's inequality to take advantage of the low variance of the $\hat{p}_i$s. For sufficiently large sample sizes, however, the high-confidence regime (discussed below) begins to take over. Finally, there is one sense in which this case is "easier" to analyze than that of known $\{p_i\}$: since the summands in (23) are bounded, Hoeffding's inequality gives nontrivial results and there is no need for more advanced tools such as the Kearns-Saul inequality (9) (which is actually inapplicable in this case).

## 4.2 High-confidence regime

In the high-confidence regime, each estimated competence $\hat{p}_i$ is close to the true value $p_i$ with high probability. To formalize this, fix some $0 < \delta < 1$, $0 < \varepsilon \leq 5$, and put $q_i = 1 - p_i$, $\hat{q}_i = 1 - \hat{p}_i$. We will set the empirical weights according to the "plug-in" Nitzan-Paroush rule

$$\hat{w}_i^{\text{HC}} := \log\frac{\hat{p}_i}{\hat{q}_i}, \qquad i \in [n], \tag{25}$$

which induces the empirical decision rule $\hat{f}^{\text{HC}}$ and raises immediate concerns about $\hat{w}_i^{\text{HC}} = \pm\infty$. We give two kinds of bounds on $\mathbb{P}(\hat{f}^{\text{HC}} \neq Y)$: nonadaptive and adaptive. In the nonadaptive analysis, we show that for $m_i \min\{p_i, q_i\} \gg 1$, with high probability $|w_i - \hat{w}_i^{\text{HC}}| \ll 1$, and thus a "perturbed" version of Theorem 1(i) holds (and in particular, $w_i^{\text{HC}}$ will be finite with high probability). In the adaptive analysis, we allow $\hat{w}_i^{\text{HC}}$ to take on infinite values[3] and show (perhaps surprisingly) that this decision rule also asymptotically achieves the rate of Theorem 1(i).

**Nonadaptive analysis.** The following result captures our analysis of the nonadaptive agent:

**Theorem 6.** *Let* $0 < \delta < 1$ *and* $0 < \varepsilon < \min\{5, 2\Phi/n\}$. *If*

$$m_i \min\{p_i, q_i\} \geq 3\left(\frac{\sqrt{4\varepsilon + 1} - 1}{4}\right)^{-2}\log\frac{4n}{\delta}, \qquad i \in [n], \tag{26}$$

*then*

$$\mathbb{P}\left(\hat{f}^{\text{HC}}(\mathbf{X}, \mathbf{K}) \neq Y\right) \leq \delta + \exp\left[-\frac{(2\Phi - \varepsilon n)^2}{8\Phi}\right]. \tag{27}$$

**Remark.** *For fixed* $\{p_i\}$ *and* $\min_{i \in [n]} m_i \to \infty$, *we may take* $\delta$ *and* $\varepsilon$ *arbitrarily small — and in this limiting case, the bound of Theorem 1(i) is recovered.*

**Adaptive analysis.** Theorem 6 has the drawback of being *nonadaptive*, in that its assumptions (26) and conclusions (27) depend on the unknown $\{p_i\}$ and hence cannot be evaluated by the agent (the bound in (24) is also nonadaptive[4]). In the *adaptive* (fully empirical) approach, all results are stated in terms of empirically observed quantities:

**Theorem 7.** *Choose any[5] $\delta \geq \sum_{i=1}^{n} \frac{1}{\sqrt{m_i}}$ and let $R$ be the event where* $\exp\left(-\frac{1}{2}\sum_{i=1}^{n}(\hat{p}_i - \frac{1}{2})\hat{w}_i^{\mathrm{HC}}\right) \leq \frac{\delta}{2}$. *Then* $\mathbb{P}\left(R \cap \left\{\hat{f}^{\mathrm{HC}}(\mathbf{X}, \mathbf{K}) \neq Y\right\}\right) \leq \delta$.

**Remark 1.** *Our interpretation for Theorem 7 is as follows. The agent observes the committee profile $\mathbf{K}$, which determines the $\{\hat{p}_i, \hat{w}_i^{\mathrm{HC}}\}$, and then checks whether the event $R$ has occurred. If not, the adaptive agent refrains from making a decision (and may choose to fall back on the low-confidence approach described previously). If $R$ does hold, however, the agent predicts $Y$ according to $\hat{f}^{\mathrm{HC}}$. Observe that the event $R$ will only occur if the empirical committee potential $\hat{\Phi} = \sum_{i=1}^{n}(\hat{p}_i - \frac{1}{2})\hat{w}_i^{\mathrm{HC}}$ is sufficiently large — i.e., if enough of the experts' competences are sufficiently far from $\frac{1}{2}$. But if this is not the case, little is lost by refraining from a high-confidence decision and defaulting to a low-confidence one, since near $\frac{1}{2}$, the two decision procedures are very similar.*

*As explained above, there does not exist a nontrivial a priori upper bound on $\mathbb{P}(\hat{f}^{\mathrm{HC}}(\mathbf{X}, \mathbf{K}) \neq Y)$ absent any knowledge of the $p_i$s. Instead, Theorem 7 bounds the probability of the agent being "fooled" by an unrepresentative committee profile.[6] Note that we have done nothing to prevent $\hat{w}_i^{\mathrm{HC}} = \pm\infty$, and this may indeed happen. Intuitively, there are two reasons for infinite $\hat{w}_i^{\mathrm{HC}}$: (a) noisy $\hat{p}_i$ due to $m_i$ being too small, or (b) the $i^{\mathrm{th}}$ expert is actually highly (in)competent, which causes $\hat{p}_i \in \{0, 1\}$ to be likely even for large $m_i$. The $1/\sqrt{m_i}$ term in the bound insures against case (a), while in case (b), choosing infinite $\hat{w}_i^{\mathrm{HC}}$ causes no harm (as we show in the proof).*

*Proof of Theorem 7.* We will write the probability and expectation operators with subscripts (such as $\mathbf{K}$) to indicate the random variable(s) being summed over. Thus,

$$\mathbb{P}_{\mathbf{K},\mathbf{X},Y}\left(R \cap \left\{\hat{f}^{\mathrm{HC}}(\mathbf{X}, \mathbf{K}) \neq Y\right\}\right) = \mathbb{P}_{\mathbf{K},\boldsymbol{\eta}}\left(R \cap \left\{\hat{\mathbf{w}}^{\mathrm{HC}} \cdot \boldsymbol{\eta} \leq 0\right\}\right)$$
$$= \mathbb{E}_{\mathbf{K}}\left[\mathbb{1}_R \cdot \mathbb{P}_{\boldsymbol{\eta}}\left(\hat{\mathbf{w}}^{\mathrm{HC}} \cdot \boldsymbol{\eta} \leq 0 \,|\, \mathbf{K}\right)\right].$$

Recall that the random variable $\boldsymbol{\eta} \in \{\pm 1\}^n$, with probability mass function $P(\boldsymbol{\eta}) = \prod_{i:\eta_i=1} p_i \prod_{i:\eta_i=-1} q_i$, is independent of $\mathbf{K}$, and hence

$$\mathbb{P}_{\boldsymbol{\eta}}\left(\hat{\mathbf{w}}^{\mathrm{HC}} \cdot \boldsymbol{\eta} \leq 0 \,|\, \mathbf{K}\right) = \mathbb{P}_{\boldsymbol{\eta}}\left(\hat{\mathbf{w}}^{\mathrm{HC}} \cdot \boldsymbol{\eta} \leq 0\right). \tag{28}$$

Define the random variable $\hat{\boldsymbol{\eta}} \in \{\pm 1\}^n$ (conditioned on $\mathbf{K}$) by the probability mass function $P(\hat{\boldsymbol{\eta}}) = \prod_{i:\eta_i=1} \hat{p}_i \prod_{i:\eta_i=-1} \hat{q}_i$, and the set $A \subseteq \{\pm 1\}^n$ by $A = \left\{\mathbf{x} : \hat{\mathbf{w}}^{\mathrm{HC}} \cdot \mathbf{x} \leq 0\right\}$. Now

$$\left|\mathbb{P}_{\boldsymbol{\eta}}\left(\hat{\mathbf{w}}^{\mathrm{HC}} \cdot \boldsymbol{\eta} \leq 0\right) - \mathbb{P}_{\hat{\boldsymbol{\eta}}}\left(\hat{\mathbf{w}}^{\mathrm{HC}} \cdot \hat{\boldsymbol{\eta}} \leq 0\right)\right| = \left|\mathbb{P}_{\boldsymbol{\eta}}(A) - \mathbb{P}_{\hat{\boldsymbol{\eta}}}(A)\right| \leq \max_{A \subseteq \{\pm 1\}^n} \left|\mathbb{P}_{\boldsymbol{\eta}}(A) - \mathbb{P}_{\hat{\boldsymbol{\eta}}}(A)\right|$$

$$= \|\mathbb{P}_{\boldsymbol{\eta}} - \mathbb{P}_{\hat{\boldsymbol{\eta}}}\|_{\mathrm{TV}} \leq \sum_{i=1}^{n} |p_i - \hat{p}_i| =: M,$$

where the last inequality follows from a standard tensorization property of the total variation norm $\|\cdot\|_{\mathrm{TV}}$, see e.g. [33, Lemma 2.2]. By Theorem 1(i), we have $\mathbb{P}_{\hat{\boldsymbol{\eta}}}\left(\hat{\mathbf{w}}^{\mathrm{HC}} \cdot \hat{\boldsymbol{\eta}} \leq 0\right) \leq \exp\left(-\frac{1}{2}\sum_{i=1}^{n}(\hat{p}_i - \frac{1}{2})\hat{w}_i^{\mathrm{HC}}\right)$, and hence $\mathbb{P}_{\boldsymbol{\eta}}\left(\hat{\mathbf{w}}^{\mathrm{HC}} \cdot \boldsymbol{\eta} \leq 0\right) \leq M + \exp\left(-\frac{1}{2}\sum_{i=1}^{n}(\hat{p}_i - \frac{1}{2})\hat{w}_i^{\mathrm{HC}}\right)$. Invoking (28), we substitute the right-hand side above into (28) to obtain

$$\mathbb{P}_{\mathbf{K},\mathbf{X},Y}\left(R \cap \left\{\hat{f}^{\mathrm{HC}}(\mathbf{X}, \mathbf{K}) \neq Y\right\}\right) \leq \mathbb{E}_{\mathbf{K}}\left[\mathbb{1}_R \cdot \left(M + \exp\left(-\frac{1}{2}\sum_{i=1}^{n}(\hat{p}_i - \tfrac{1}{2})\hat{w}_i^{\mathrm{HC}}\right)\right)\right]$$

$$\leq \mathbb{E}_{\mathbf{K}}[M] + \mathbb{E}_{\mathbf{K}}\left[\mathbb{1}_R \exp\left(-\frac{1}{2}\sum_{i=1}^{n}(\hat{p}_i - \tfrac{1}{2})\hat{w}_i^{\mathrm{HC}}\right)\right].$$

By the definition of $R$, the second term on the last right-hand side is upper-bounded by $\delta/2$. To estimate $M$, we invoke a simple mean absolute deviation bound (cf. [34]):

$$\mathbb{E}_{\mathbf{K}}|p_i - \hat{p}_i| \leq \sqrt{\frac{p_i(1 - p_i)}{m_i}} \leq \frac{1}{2\sqrt{m_i}},$$

which finishes the proof. $\qquad\qquad\qquad\qquad\qquad\qquad\qquad\qquad\qquad\qquad\qquad\qquad\qquad\quad$ $\square$

**Remark.** *The improvement mentioned in Footnote 5 is achieved via a refinement of the bound* $\|\mathbb{P}_{\boldsymbol{\eta}} - \mathbb{P}_{\hat{\boldsymbol{\eta}}}\|_{\mathrm{TV}} \leq \sum_{i=1}^{n} |p_i - \hat{p}_i|$ *to* $\|\mathbb{P}_{\boldsymbol{\eta}} - \mathbb{P}_{\hat{\boldsymbol{\eta}}}\|_{\mathrm{TV}} \leq \alpha\left(\{|p_i - \hat{p}_i| : i \in [n]\}\right)$, *where* $\alpha(\cdot)$ *is the function defined in [33, Lemma 4.2].*

**Open problem.** As argued in Remark 1, Theorem 7 achieves the optimal asymptotic rate in $\{p_i\}$. Can the dependence on $\{m_i\}$ be improved, perhaps through a better choice of $\hat{\mathbf{w}}^{\mathrm{HC}}$?

## 5 Unknown competences: Bayesian

A shortcoming of Theorem 7 is that when condition $R$ fails, the agent is left with no estimate of the error probability. An alternative (and in some sense cleaner) approach to handling unknown expert competences $p_i$ is to assume a known prior distribution over the competence levels $p_i$. The natural choice of prior for a Bernoulli parameter is the Beta distribution, namely $p_i \sim \mathrm{Beta}(\alpha_i, \beta_i)$ with density $\frac{p_i^{\alpha_i - 1} q_i^{\beta_i - 1}}{B(\alpha_i, \beta_i)}$, where $\alpha_i, \beta_i > 0$, $q_i = 1 - p_i$ and $B(x, y) = \Gamma(x)\Gamma(y)/\Gamma(x + y)$. Our full probabilistic model is as follows. Each of the $n$ expert competences $p_i$ is drawn independently from a Beta distribution with known parameters $\alpha_i, \beta_i$. Then the $i^{\mathrm{th}}$ expert, $i \in [n]$, is queried independently $m_i$ times, with $k_i$ correct predictions and $m_i - k_i$ incorrect ones. As before, $\mathbf{K} = (k_1, \ldots, k_n)$ is the (random) committee profile. Absent direct knowledge of the $p_i$s, the agent relies on an empirical decision rule $\hat{f} : (\mathbf{x}, \mathbf{k}) \mapsto \{\pm 1\}$ to produce a final decision based on the expert inputs $\mathbf{x}$ together with the committee profile $\mathbf{k}$. A decision rule $\hat{f}^{\mathrm{Ba}}$ is *Bayes-optimal* if it minimizes $\mathbb{P}(\hat{f}(\mathbf{X}, \mathbf{K}) \neq Y)$, which is formally identical to (18) but semantically there is a difference: the former is over the $p_i$ in addition to $(\mathbf{X}, Y, \mathbf{K})$. Unlike the frequentist approach, where no optimal empirical decision rule was possible, the Bayesian approach readily admits one: $\hat{f}^{\mathrm{Ba}}(\mathbf{x}, \mathbf{k}) = \mathrm{sign}\left(\sum_{i=1}^{n} \hat{w}_i^{\mathrm{Ba}} x_i\right)$, where

$$\hat{w}_i^{\mathrm{Ba}} = \log \frac{\alpha_i + k_i}{\beta_i + m_i - k_i}. \tag{29}$$

Notice that for $0 < p_i < 1$, we have $\hat{w}_i^{\mathrm{Ba}} \xrightarrow[m_i \to \infty]{} w_i$, almost surely, both in the frequentist and the Bayesian interpretations. Unfortunately, although $\mathbb{P}(\hat{f}^{\mathrm{Ba}}(\mathbf{X}, \mathbf{K}) \neq Y) = \mathbb{P}(\hat{\mathbf{w}}^{\mathrm{Ba}} \cdot \boldsymbol{\eta} \leq 0)$ is a deterministic function of $\{\alpha_i, \beta_i, m_i\}$, we are unable to compute it at this point, or even give a non-trivial bound. The main source of difficulty is the coupling between $\hat{\mathbf{w}}^{\mathrm{Ba}}$ and $\boldsymbol{\eta}$.

**Open problem.** Give a non-trivial estimate for $\mathbb{P}(\hat{f}^{\mathrm{Ba}}(\mathbf{X}, \mathbf{K}) \neq Y)$.

## 6 Discussion

The classic and seemingly well-understood problem of the consistency of weighted majority votes continues to reveal untapped depth and suggest challenging unresolved questions. We hope that the results and open problems presented here will stimulate future research.

## Footnotes

[1] Without loss of generality, ties are considered to be errors.

[2] For $m_i \min\{p_i, q_i\} \ll 1$, the estimated competences $\hat{p}_i$ may well take values in $\{0, 1\}$, in which case $\log(\hat{p}_i/\hat{q}_i) = \pm\infty$. The rule in (21) is essentially a first-order Taylor approximation to $w(\cdot)$ about $p = \frac{1}{2}$.

[3] When the decision rule is faced with evaluating sums involving $\infty - \infty$, we automatically count this as an error.

[4]The term *oracle* was suggested by a referee for this setting.

[5] Actually, as the proof will show, we may take $\delta$ to be a smaller value, but with a more complex dependence on $\{m_i\}$, which simplifies to $2[1 - (1 - (2\sqrt{m})^{-1})^n]$ for $m_i \equiv m$.

[6]These adaptive bounds are similar in spirit to *empirical Bernstein* methods, [30, 31, 32], where the agent's confidence depends on the empirical variance.

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
