[Reviews · NeurIPS 2014]

Submitted by Assigned_Reviewer_20

This paper considers weighted majority algorithm and establishes consistency (error rate of the aggregator tending to zero) results under two settings: (1) when the competence level (risk of each expert) is known in advance and (2) when it is estimated. For case (2), frequentist and Bayesian methods for estimating the competence level are provided.

For case (1), consistency is established in terms of providing upper and lower bounds on the error rate of the aggregator, which involve standard calculations ( apart from the fact that upper bound is established by invoking a result by Kearns and Saul, instead of Hoeffding's inequality). For case (2) under the frequentist setting, an independent set of labeled inputs is used to estimate the competence level of each expert. Depending on the confidence level, two estimates (and corresponding analyses) are provided. Under Bayesian setting, the estimator proposed is more or less standard and the analysis is entirely left as an open problem.

The main problem with the paper is the novelty aspect of the work. The estimators provided and the analyses provided are based on standard tools and there is no 'novel theoretical contribution' as such. From a practical perspective, assuming access to labeled samples in order to estimate the competence level serves as a drawback compared to the Dawid-Skene method, using which both competence level and ground truth could be estimated without labels (apart from the references provided in the paper, two related references are [1] and [2] below). But still, the problem considered in the paper is very relevant (especially due to several crowd-sourcing applications). An advantage of weighted majority method over the Dawid-Skene model, is the former makes no modeling assumptions. Hence a theoretical analysis characterizing conditions when weighted majority performs better compared to the David-Skene method would make the paper much stronger.

[1] http://jmlr.org/papers/v11/donmez10a.html
[2] http://arxiv.org/abs/1310.5764
Summary: This paper considers weighted majority algorithm and establishes consistency (error rate of the aggregator tending to zero) results. Drawback of the paper is the novelty aspect - there is no novel theoretical contribution as such or particularly striking practical implications of the analysis provided.

Submitted by Assigned_Reviewer_22

This paper considers the problem of combining experts' guesses in a
classification setting. Unlike the sequential, adversarial setting,
the problem is based on static guesses with known or estimated
accuracies (competencies). In particular, the authors show conditions
that guarantee when the Nitzan-Paroush weighted majority rule is
consistent.

The paper is well-written and very interesting. The results are very
clean and rigorous. The authors provide lots of intuition and even
point out interesting open problems along the way. Interestingly,
standard concentration of measure results like Hoeffding's inequality
and Bernstein's inequality are not adequate. Instead, less known but
sharper bounds are used.
Summary: Clear, interesting analysis if the weighted majority rule
in the non-sequential setting.

Submitted by Assigned_Reviewer_30

This paper analyzes the Nitzan-Paroush strategy for assigning weights to conditionally (on the outcome) independent experts, which is optimal in expectation. The authors provide exponential concentration inequalities (both upper and lower bounds on probabilities) for the error of this optimal rule. The upper bounds are sharp and derived based on the Kearns-Saul inequality. They continue to analyze the situation where the quality of the experts is not given and must be estimated from data, giving frequentist procedures (which are analyzed) and Bayesian procedures (which are not analyzed).

QUALITY

The results are interesting, nontrivial and appear correct, as far as I checked. I was struck with the use of the Kearns-Saul inequality, which (although I have seen it before and know it was designed for other goals) seems almost magically suitable for application in this problem. The exponential bound Theorem 1(i) is quite strong, being linear in the nr of experts who are correct with probability > 1/2 + epsilon, for fixed epsilon > 0.

I do have some questions/small issues about the estimators, both the frequentist and the Bayesian ones.

First, as a very minor point, I'd rather call 'adaptive' 'fully empirical' and 'nonadaptive' 'oracle' by the way, that seems to more correctly describe the difference.

Second:
Theorem 7: adaptive, high-confidence, frequentist case:

This theorem is only useful if the authors can show that the event R will hold with large probability for sufficiently large sample sizes. (Otherwise it may be that we always have to refrain from making a high-confidence decision, making the results useless).
The authors should say some more about the sample sizes when we can expect R to hold, given the (oracle, nonadaptive) probabilities
(so that 'if we are lucky about the real probabilities, then with high probability we get a situation in which the adaptive bound is useful')

About the Bayesian method: this seems to be based on using the standard, unconditional Naive Bayes model in which the experts opinions are viewed as X-data and the outcome as Y. But I guess that most Bayesians would condition on the X's and use conditional likelihood - and then Naive Bayes becomes logistic regression, which often works better anyway. The authors should say something about this additional possibility. Is there any reason why they have not considered it?

Why naive Bayes and not logistic regression approach?

CLARITY

The paper is quite well-written; I am not an expert in the probabilistic weighted-majority analysis but had no trouble following the paper.

ORIGINALITY

The results are somewhat original, the *proof techniques* are very original.

SIGNIFICANCE

Reasonably high, esp. given the use of the Kearns-Saul inequality.
Summary: The authors give concentration bounds on the error of weighted majority voting with optimal weights, and show how to learn the weights from data. Results are interesting, proof technique even more so.
Author Feedback
Author rebuttal: We thank the referees for detailed and helpful comments.

R_20:
"The main problem with the paper is the novelty aspect of the work. The estimators
provided and the analyses provided are based on standard tools and there is no
'novel theoretical contribution' as such."
We respectfully disagree. For the upper bound in Theorem 1, the use of the Kearns-Saul
inequality appears to be quite novel in this setting. This is particularly relevant
since the standard bounds (such as Chernoff-Hoeffding or Bennett-Bernstein break down).
The lower bound likewise employs a technique that we have not seen elsewhere.

"two related references are [1] and [2] below"
We thank the referee for these quite relevant references, and will be sure to discuss them
in the revision.

R_22,R_30:
We deeply appreciate the encouraging remarks.

R_30: "The exponential bound Theorem 1(i) is quite strong,
being linear in the nr of experts who are correct with probability > 1/2 + epsilon, for
fixed epsilon > 0."
We daresay it's even stronger -- since a single highly competent expert is sufficient to
achieve an arbitrarily small error probability.

"First, as a very minor point, I'd rather call 'adaptive' 'fully empirical' and
'nonadaptive' 'oracle' by the way, that seems to more correctly describe the
difference."
Thank you for this suggestion, we will implement it in the revision.

"Theorem 7: adaptive, high-confidence, frequentist case:
This theorem is only useful if the authors can show that the event R will hold with
large probability for sufficiently large sample sizes. (Otherwise it may be that we
always have to refrain from making a high-confidence decision, making the results
useless)."
Indeed, the event R will hold when the estimated p_i's are "favorable" in the sense of
inducing a large empirical committee profile. But if they are not (i.e., if many of
them are close to 1/2), then there is little lost by refraining from a high-confidence
decision and defaulting to a low-confidence one, since near 1/2, the two decision procedures
are very similar. We will elaborate on this in the revision.

"The authors should say some more about the sample sizes when we can expect R
to hold, given the (oracle, nonadaptive) probabilities
(so that 'if we are lucky about the real probabilities, then with high probability we
get a situation in which the adaptive bound is useful')"
Indeed, we will include a discussion in the revision.

"About the Bayesian method: this seems to be based on using the standard,
unconditional Naive Bayes model in which the experts opinions are viewed as data
and the outcome as Y. But I guess that most Bayesians would condition on
the X's and use conditional likelihood - and then Naive Bayes becomes logistic
regression, which often works better anyway. The authors should say something
about this additional possibility. Is there any reason why they have not considered
it?
Why naive Bayes and not logistic regression approach?"
This is a very interesting suggestion, which we will surely pursue (perhaps in
a separate paper).